# Understanding the Relation Between Maximum-Entropy Inverse Reinforcement Learning and Behaviour Cloning

**Seyed Kamyar Seyed Ghasemipour**
University of Toronto, Vector Institute
kamyar@cs.toronto.edu

**Shixiang Gu**
Google Brain
shanegu@google.com

**Richard Zemel**
University of Toronto, Vector Institute
zemel@cs.toronto.edu

## Abstract

In many settings, it is desirable to learn decision-making and control policies through learning or bootstrapping from expert demonstrations. The most common approaches under this framework are Behaviour Cloning (BC), and Inverse Reinforcement Learning (IRL). Recent methods for IRL have demonstrated the capacity to learn effective policies with access to a very limited set of demonstrations, a scenario in which BC methods often fail. Unfortunately, directly comparing the algorithms for these methods does not provide adequate intuition for understanding this difference in performance. This is the motivating factor for our work. We begin by presenting $f$-MAX, a generalization of AIRL (Fu et al., 2018), a state-of-the-art IRL method. $f$-MAX provides grounds for more directly comparing the objectives for LfD. We demonstrate that $f$-MAX, and by inhertance AIRL, is a subset of the cost-regularized IRL framework laid out by Ho & Ermon (2016). We conclude by empirically evaluating the factors of difference between various LfD objectives in the continuous control domain.

## 1 Introduction

Modern advances in reinforcement learning aim to alleviate the need for hand-engineered decision-making and control algorithms by designing general purpose methods that learn to optimize provided reward functions. In many cases however, it is either too challenging to optimize a given reward (e.g. due to sparsity of signal), or it is simply impossible to design a reward function that captures the intricate details of desired outcomes. One approach to overcoming such hurdles is Learning from Demonstrations (LfD), where algorithms are provided with expert demonstrations of how to accomplish desired tasks.

The most common approaches in the LfD framework are Behaviour Cloning (BC) and Inverse Reinforcement Learning (IRL) (Russell, 1998; Ng et al., 2000). In standard BC, learning from demonstrations is treated as a supervised learning problem and policies are trained to regress expert actions from a dataset of expert demonstrations. Other forms of Behaviour Cloning, such as DAgger (Ross et al., 2011), consider how to make use of an expert in a more optimal fashion. On the other hand, in IRL the aim is to infer the reward function of the expert, and subsequently train a policy to optimize this reward. The motivation for IRL stems from the intuition that the reward function is the most concise and portable representation of a task (Ng et al., 2000; Abbeel & Ng, 2004).

Unfortunately, the standard IRL formulation (Ng et al., 2000) faces degeneracy issues [1]. A successful framework for overcoming such challenges is the Maximum-Entropy (Max-Ent) IRL method (Ziebart et al., 2008; Ziebart, 2010). A line of research stemming from the Max-Ent IRL framework has lead to recent "adversarial" methods (Ho & Ermon, 2016; Finn et al., 2016a; Fu et al.,

---

[1]for example, any policy is optimal for the constant reward function $r(s, a) = 0$

2018). These approaches aim to directly recover the expert policy ensuing from the Max-Ent IRL process without explicitly modelling the reward function, and have shown tremendous success. In benchmarks for continuous control (Brockman et al., 2016), these methods outperform Behaviour Cloning by a wide margin, particularly in the low data regime where a very limited number of expert trajectories are available.

However, it is not immediately clear why adversarial Max-Ent IRL methods would outperform BC, since at optimality both methods exactly recover the expert policy. This questions motivates the work presented here. Drawing upon the literature on $f$-divergences (Lin, 1991; Nowozin et al., 2016), we begin by presenting $f$-MAX, an algorithm for Max-Ent IRL. We demonstrate how $f$-MAX generalizes AIRL (Fu et al., 2018), and provides new intuition for what this algorithm accomplishes. Specifically we demonstrate that the objective in AIRL is equivalent to minimizing the reverse KL divergence between the joint state-action marginal distribution of the policy and that of the expert. Additionally, we demonstrate that $f$-MAX, and by inhertance AIRL, is a subset of the cost-regularized Max-Ent IRL framework laid out by Ho & Ermon (2016). From these findings, we generate hypotheses for why direct Max-Ent IRL methods outperform BC, and empirically evaluate them in continuous control benchmarks. To tease apart the differences between standard BC and AIRL, we also address the degeneracy of $f$-MAX in a special case, and provide a one-line modification of AIRL, named FAIRL, which minimizes the *forward* KL divergence between the joint state-action marginal of the expert and the policy. We discuss how our findings may relate to common observations (Bishop, 2006) regarding the mode-covering/mode-seeking behaviour of the different KL divergence directions.

## 2 BACKGROUND

### 2.1 MAXIMUM ENTROPY INVERSE REINFORCEMENT LEARNING

Consider a Markov Decision Process (MDP) represented as a tuple $(\mathcal{S}, \mathcal{A}, \mathcal{P}, r, \rho_0, \gamma)$ with state-space $\mathcal{S}$, action-space $\mathcal{A}$, dynamics $\mathcal{P} : \mathcal{S} \times \mathcal{A} \times \mathcal{S} \rightarrow [0, 1]$, reward function $r(s, a)$, initial state distribution $\rho_0$, and discount factor $\gamma \in (0, 1)$. In Maximum Entropy (Max-Ent) reinforcement learning (Todorov, 2008; Toussaint, 2009; Rawlik et al., 2013; Fox et al., 2015; Haarnoja et al., 2017; 2018), the goal is to find a policy $\pi$ such that trajectories sampled using this policy follow the distribution

$$p(\tau) = \frac{1}{Z}\exp(R(\tau)) \tag{1}$$

where $\tau = (s_0, a_0, s_1, a_1, ...)$ denotes a trajectory, and $R(\tau) = \sum_t r(s_t, a_t)$ and $Z$ is the partition function. Hence, trajectories that accumulate more reward are exponentially more likely to be sampled.

Converse to the standard RL setting, in Max-Ent IRL (Ziebart et al., 2008; Ziebart, 2010) we are instead presented with an optimal policy $\pi^{exp}$, or more realistically, sample trajectories from such a policy, and we seek to find a reward function $r$ that maximizes the likelihood of the trajectories sampled from $\pi^{exp}$. Formally, our objective is:

$$\max_r \mathbb{E}_{\tau \sim \pi^{exp}}[R(\tau) - \log Z] \tag{2}$$

Being an energy-based modelling objective, the difficulty in performing this optimization arises from estimating the partition function $Z$. Initial methods addressed this problem using dynamic programming (Ziebart et al., 2008; Ziebart, 2010), and recent approaches present methods aimed at intractable domains with unknown dynamics (Finn et al., 2016b; Ho & Ermon, 2016; Finn et al., 2016a; Fu et al., 2018; Kostrikov et al., 2018).

Instead of recovering the expert's reward function and policy, recent successful methods in Max-Ent IRL aim to directly recover the policy that would result from the full process. Since such methods only recover the policy, it would be more accurate to refer to them as Imitation Learning algorithms. However, to avoid confusion with Behaviour Cloning methods, in this work we will refer to them as *direct* methods for Max-Ent IRL.

**GAIL: Generative Adversarial Imitation Learning**   Before describing the work of Ho & Ermon (2016), we establish the definition of causal entropy $\mathcal{H}^{\text{causal}}(\pi) := \mathbb{E}_{\rho^\pi(s)} [\log \pi(a|s)]$ (Ziebart, 2010; Bloem & Bambos, 2014). Intuitively, causal entropy can be thought of as the "amount of options" the policy has in each state, in expectation.

Let $\mathcal{C}$ denote a class of cost functions (negative reward functions). Furthermore, let $\rho^{\text{exp}}(s, a), \rho^\pi(s, a)$ denote the state-action marginal distributions of the expert and student policy respectively. Ho & Ermon (2016) begin with a regularized Max-Ent IRL objective,

$$\text{IRL}_\psi(\pi^{\text{exp}}) := \underset{c \in \mathcal{C}}{\arg \max} -\psi(c) + \left( \min_\pi -\mathcal{H}^{\text{causal}}(\pi) + \mathbb{E}_{\rho^\pi(s,a)} [c(s,a)] \right) - \mathbb{E}_{\rho^{\text{exp}}(s,a)} [c(s,a)]$$

(3)

where $\psi : \mathcal{C} \to \mathbb{R}$ is a convex regularization function on the space of cost functions, and $\text{IRL}_\psi(\pi^{\text{exp}})$ returns the optimal cost function given the expert and choice of regularization. Also, while not immediately clear, note that $\min_\pi -\mathcal{H}^{\text{causal}}(\pi) + \mathbb{E}_\pi [c(s,a)]$ is the Max-Ent RL objective given cost function $c(s,a)$. Let $\text{RL}(c) := \arg \min_\pi -\mathcal{H}^{\text{causal}}(\pi) + \mathbb{E}_\pi [c(s,a)]$, be a function that returns the optimal Max-Ent policy given cost $c(s,a)$. Ho & Ermon (2016) show that

$$\text{RL} \circ \text{IRL}_\psi(\pi^{\text{exp}}) = \arg \min_\pi -\mathcal{H}^{\text{causal}}(\pi) + \psi^* (\rho^\pi(s,a) - \rho^{\text{exp}}(s,a))$$

(4)

where $\psi^*$ denotes the convex conjugate of $\psi$. This tells us that if we were to find the cost function $c(s,a)$ using the regularized Max-Ent IRL objective 3, and subsequently find the optimal Max-Ent policy for this cost, we would arrive at the same policy had we directly optimized objective 4 by searching for the policy.

Directly optimizing 4 is challenging for many choices of $\psi$. Interestingly however, Ho & Ermon (2016) show that for any symmetric $f$-divergences (Lin, 1991), there exists a choice of $\psi$ such that equation 4 is equivalent to $\text{RL} \circ \text{IRL}_\psi(\pi^{\text{exp}}) = \arg \min_\pi \mathcal{H}^{\text{causal}}(\pi) + D_f (\rho^\pi(s,a) || \rho^{\text{exp}}(s,a))$. In such settings, due to a close connection between binary classifiers and symmetric $f$-divergences (Nguyen et al., 2009), efficient algorithms can be formed.

The special case for Jensen-Shannon divergence leads to the successful method dubbed Generative Adversarial Imitation Learning (GAIL). As before, let $\rho^{\text{exp}}(s, a), \rho^\pi(s, a)$ denote the state-action marginal distributions of the expert and student policy respectively. Let $D(s, a) : \mathcal{S} \times \mathcal{A} \to [0, 1]$ be a binary classifier - often referred to as the discriminator - for identifying positive samples (sampled from $\rho^{\text{exp}}(s, a)$) from negative samples (sampled from $\rho^\pi(s, a)$). Using RL, the student policy is trained to maximize $\mathbb{E}_{\tau \sim \pi} [\sum_t \log D(s_t, a_t)] - \lambda \mathcal{H}^{\text{causal}}(\pi)$, where $\lambda$ is a hyperparameter. The training procedure alternates between optimizing the discriminator and updating the policy. As noted, it is shown that this training procedure minimizes the Jensen-Shannon divergence between $\rho^{\text{exp}}(s, a)$ and $\rho^\pi(s, a)$ (Ho & Ermon, 2016).

**AIRL: Adversarial Inverse Reinforcement Learning**   Subsequent to the advent of GAIL (Ho & Ermon, 2016), Finn et al. (2016a) present a theoretical discussion relating Generative Adversarial Networks (GANs) (Goodfellow et al., 2014), IRL, and energy-based models. They demonstrate how an adversarial training approach could recover the Max-Ent reward function and simultaneously train the Max-Ent policy corresponding to that reward. Building on this discussion, Fu et al. (2018) present a practical implementation of this method, named Adversarial Inverse Reinforcement Learning (AIRL).

As before, let $\rho^{\text{exp}}(s, a), \rho^\pi(s, a)$ denote the state-action marginal distributions of the expert and student policy respectively and let $D(s, a) : \mathcal{S} \times \mathcal{A} \to [0, 1]$ be the discriminator. In AIRL, the discriminator is parameterized as,

$$D(s, a) := \frac{\exp(f(s,a))}{\exp(f(s,a)) + \pi(a|s)}$$

(5)

where $f(s, a) : \mathcal{S} \times \mathcal{A} \to \mathbb{R}$, and $\pi(a|s)$ denotes the likelihood of the action under the policy. AIRL defines the reward function, $r(s, a) := \log D(s, a) - \log (1 - D(s, a))$, and sets the objective for the student policy to be the RL objective, $\max_\pi \mathbb{E}_{\tau \sim \pi} [\sum_t r(s_t, a_t)]$. As in GAIL, this leads to an iterative optimization process alternating between optimizing the discriminator and the policy.

At convergence, the advantage function of the expert is recovered. Given this observation, important considerations are made regarding how to extract the true reward function from $f(s, a)$. When the

objective is only to perform Imitation Learning, and we do not care to recover the reward function, the discriminator does not use the special parameterization discussed above and is instead direclty represented as a function $D(s, a) : \mathcal{S} \times \mathcal{A} \to [0, 1]$, as done in GAIL (Ho & Ermon, 2016).

**Performance With Respect to BC**   Methods such as GAIL and AIRL have demonstrated significant performance gains compared to Behaviour Cloning. In particular, in standard Mujoco benchmarks (Todorov et al., 2012; Brockman et al., 2016), adversarial methods for Max-Ent IRL achieve strong performance using a *very limited* amount of demonstrations from an expert policy, an important failure scenario for standard Behaviour Cloning.

## 2.2   $f$-DIVERGENCES

Ho & Ermon (2016) demonstrate that Max-Ent IRL is the dual problem of matching $\rho^{\pi}(s, a)$ to $\rho^{\exp}(s, a)$; indeed as noted above, GAIL (Ho & Ermon, 2016) optimizes the Jensen-Shannon divergence between the two distributions. In section 3 we present $f$-MAX, a method for matching $\rho^{\pi}(s, a)$ to $\rho^{\exp}(s, a)$ using arbitrary $f$-divergences (Lin, 1991). Hence, in this section we recall this class of statistical divergences as well as methods for using them for training generative models.

Let $P, Q$ be two distributions with density functions $p, q$. For any convex, lower-semicontinuous function $f : \mathbb{R}^+ \to \mathbb{R}$ a statistical divergence can be defined as: $D_f(P||Q) = \int_{\chi} q(x) f\left(\frac{p(x)}{q(x)}\right)$. Divergences derived in this manner are called *f-divergences* and amongst many interesting divergences, include the forward and reverse KL.

Nguyen et al. (2010) present a variational estimation method for $f$-divergences between arbitrary distributions P, Q. Using the notation of Nowozin et al. (2016) we can write,

$$D_f(P||Q) \geq \sup_{T_\omega \in \mathcal{T}} \left( \mathbb{E}_{x \sim P} \left[ T_\omega(x) \right] - \mathbb{E}_{x \sim Q} \left[ f^*(T_\omega(x)) \right] \right) \tag{6}$$

where $\mathcal{T}$ is an arbitrary class of functions $T_\omega : X \to \mathbb{R}$, and $f^*$ is the convex conjugate of $f$. Under mild conditions (Nguyen et al., 2010) equality holds between the two sides. Motivated by this variational approximation as well as Generative Adversarial Networks (GANs) (Goodfellow et al., 2014), Nowozin et al. (2016) present an iterative optimization scheme for matching an implicit distribution[2] Q to a fixed distribution P using any $f$-divergence. For a given $f$-divergence, the corresponding minimax optimization is,

$$\min_Q \max_{T_\omega} F(\theta, \omega) = \mathbb{E}_{x \sim P} \left[ T_\omega(x) \right] - \mathbb{E}_{x \sim Q} \left[ f^*(T_\omega(x)) \right] \tag{7}$$

Nowozin et al. (2016) discuss practical parameterizations of $T_\omega$, but to avoid notational clutter we will use the form above.

## 3   $f$-MAX: $f$-DIVERGENCE MAX-ENT IRL

We begin by presenting $f$-MAX, a generalization of AIRL (Fu et al., 2018) which provides a more intuitive interpretation of what similar algorithms accomplish.

Imagine, for some $f$, we aim to train a policy by optimizing the $f$-divergence $D_f\left(\rho^{\exp}(s, a)||\rho^{\pi}(s, a)\right)$. To do so, we propose the following iterative optimization procedure,

$$\max_{T_\omega} \mathbb{E}_{(s,a) \sim \rho^{\exp}(s,a)} \left[ T_\omega(s, a) \right] - \mathbb{E}_{(s,a) \sim \rho^{\pi}(s,a)} \left[ f^*(T_\omega(s, a)) \right] \tag{8}$$

$$\max_{\pi} \mathbb{E}_{\tau \sim \pi} \left[ \sum_t f^*(T_\omega(s_t, a_t)) \right] \tag{9}$$

---

[2]We use the term "implicit distributions" to refer to distributions we can efficiently sample from, e.g. GAN (Goodfellow et al., 2016) generators

where $f^*$ and $T_\omega$ are as defined in section 2.2. Equation 8 is the same as the inner maximization of the $f$-GAN objective in equation 7; this objective optimizes $T_\omega$ so that equation 8 best approximates $D_f\left(\rho^{\text{exp}}(s,a)||\rho^\pi(s,a)\right)$.

On the other hand, for the policy objective, using the identities in appendix A we have,

$$\frac{1}{T}\mathbb{E}_{\tau\sim\pi}\left[\sum_t f^*(T_\omega^\pi(s_t,a_t))\right] \propto \mathbb{E}_{(s,a)\sim\rho^\pi(s,a)}\left[f^*(T_\omega^\pi(s,a))\right] \tag{10}$$

which implies that the policy objective is equivalent to minimizing equation 8 with respect to $\pi$. With an identical proof as in Goodfellow et al. (2014, Proposition 2), if in each iteration the optimal $T_\omega$ is found, the described optimization procedure converges to the global optimum where the policy's state-action marginal distribution matches that of the expert's. This is equivalent to iteratively computing $D_f\left(\rho^{\text{exp}}(s,a)||\rho^\pi(s,a)\right)$ and optimizing the policy to minimize it.

### 3.1 COROLLARY: A SIMPLE DERIVATION AND INTUITION FOR AIRL

Choosing $f(u) := -\log u$ leads to $D_f(\rho^{\text{exp}}(s,a)||\rho^\pi(s,a)) = \text{KL}(\rho^\pi(s,a)||\rho^{\text{exp}}(s,a))$. This divergence is commonly referred to as the "reverse" KL divergence. In this setting we have, $f^*(t) = -1 - \log(-t)$, and $T_\omega^\pi(s,a) = -\frac{\rho^\pi(s,a)}{\rho^{\text{exp}}(s,a)}$ (Nowozin et al., 2016). Hence, given $T_\omega^\pi$, the policy objective in equation 9 takes the form,

$$\max_\pi \mathbb{E}_{\tau\sim\pi}\left[\sum_t f^*(T_\omega^\pi(s_t,a_t))\right] = \max_\pi \mathbb{E}_{\tau\sim\pi}\left[\sum_t \log\rho^{\text{exp}}(s_t,a_t) - \log\rho^\pi(s_t,a_t) - 1\right] \tag{11}$$

On the other hand, plugging the optimal discriminator $D^\pi(s,a) = \frac{\rho^{\text{exp}}(s,a)}{\rho^{\text{exp}}(s,a)+\rho^\pi(s,a)}$ (Goodfellow et al., 2014) into the AIRL (Fu et al., 2018) policy objective, we get,

$$\max_\pi \mathbb{E}_{\tau\sim\pi}\left[\sum_t \log D^\pi(s_t,a_t) - \log(1-D^\pi(s_t,a_t))\right] = \mathbb{E}_{\tau\sim\pi}\left[\sum_t \log\rho^{\text{exp}}(s_t,a_t) - \log\rho^\pi(s_t,a_t)\right] \tag{12}$$

As can be seen, the right hand side of equation 12 matches that of equation 11 up to a constant [3], meaning that AIRL is solving the Max-Ent IRL problem by minimizing the reverse KL divergence, $\text{KL}(\rho^\pi(s,a)||\rho^{\text{exp}}(s,a))$!

### 3.2 RELATION TO GAIL

As discussed above, Ho & Ermon (2016) present a class of methods for Max-Ent IRL that directly retrieve the expert policy without explicitly finding the reward function of the expert (sec. 2.1). Using an interesting connection between surrogate cost functions for binary classification and $f$-divergences (Nguyen et al., 2009), Ho & Ermon (2016) derive a special case of their method for minimizing any *symmetric*[4] $f$-divergence between $\rho^{\text{exp}}(s,a)$ and $\rho^\pi(s,a)$. Choosing the symmetric $f$-divergence to be the Jensen-Shannon divergence leads to the successful special case, GAIL (sec 2.1).

Surprisingly, we now show that $f$-MAX is a subset of the cost-regularized Max-Ent IRL framework laid out in Ho & Ermon (2016)! Recall the following equations from this framework,

$$\text{IRL}_\psi(\pi^{\text{exp}}) := \arg\max_{c\in\mathcal{C}} -\psi(c) + \left(\min_\pi -\mathcal{H}^{\text{causal}}(\pi) + \mathbb{E}_{\rho^\pi(s,a)}[c(s,a)]\right) - \mathbb{E}_{\rho^{\text{exp}}(s,a)}[c(s,a)] \tag{13}$$

$$\text{RL}\circ\text{IRL}_\psi(\pi^{\text{exp}}) = \arg\min_\pi -\mathcal{H}^{\text{causal}}(\pi) + \psi^*\left(\rho^\pi(s,a) - \rho^{\text{exp}}(s,a)\right) \tag{14}$$

---

[3]In both settings of fixed finite horizon, and infinite horizon with constant probability of termination, the additional term resulting from the $-1$ is a constant.

[4]We call an $f$-divergence symmetric if for any P,Q we have $D_f(P||Q) = D_f(Q,P)$

| Method | Optimized Objective (Minimization) |
|---|---|
| Standard Behaviour Cloning | $\mathbb{E}_{\rho^{\exp}(s)}\left[\mathrm{KL}\left(\rho^{\exp}(a|s)||\rho^{\pi}(a|s)\right)\right] = -\mathbb{E}_{\rho^{\exp}(s,a)}\left[\log \rho^{\pi}(a|s)\right] + C$ |
| DAgger (Ross et al., 2011) | $\mathbb{E}_{\rho^{\mathrm{agg}_{1:n}}(s)}\left[\mathrm{KL}\left(\rho^{\exp}(a|s)||\rho^{\pi}(a|s)\right)\right]$ at iteration $n+1$ |
| FAIRL (this work, section 5) | $\mathrm{KL}(\rho^{\exp}(s,a)||\rho^{\pi}(s,a)) = -\mathbb{E}_{\rho^{\exp}(s,a)}\left[\log \rho^{\pi}(s,a)\right] - \mathcal{H}(\rho^{\exp}(s,a))$ |
| AIRL (Fu et al., 2018) | $\mathrm{KL}(\rho^{\pi}(s,a)||\rho^{\exp}(s,a)) = -\mathbb{E}_{\rho^{\pi}(s,a)}\left[\log \rho^{\exp}(s,a)\right] - \mathcal{H}(\rho^{\pi}(s,a))$ |
| GAIL (Ho & Ermon, 2016) | $D_{\mathrm{JS}}(\rho^{\exp}(s,a)||\rho^{\pi}(s,a)) - \lambda\mathcal{H}^{\mathrm{causal}}(\pi)$ |
| Ho & Ermon (2016) symm. $f$-div | $D_{f\text{-symm}}(\rho^{\pi}(s,a)||\rho^{\exp}(s,a)) - \lambda\mathcal{H}^{\mathrm{causal}}(\pi)$ |
| $f$-MAX (this work, section 3) | $D_{f}(\rho^{\pi}(s,a)||\rho^{\exp}(s,a))$ |

Table 1: The objective function for various imitation learning algorithms, written in a common form as the minimization of statistical divergences. $\mathcal{H}(\cdot)$ denotes entropy, $\mathcal{H}^{\mathrm{causal}}(\pi)$ denotes the causal entropy of the policy (Ziebart, 2010; Ho & Ermon, 2016), and $\lambda$ is a hyperparameter. JS denotes the Jensen-Shannon divergence and $D_f$ indicates any $f$-divergence. For DAgger, we are showing the objective for the simplest form of the algorithm, where $\pi^{(i)}$ is the policy obtained at iteration $i$, $\pi^{(1)}$ is the expert, and $\rho^{\mathrm{agg}_{1:n}}(s) = \frac{1}{n}\sum_{i=1}^{n}\rho^{\pi^{(i)}}(s)$.

where $\psi(c) : \mathcal{C} \to \mathbb{R}$ was a closed, proper, and convex regularization function on the space of cost function, and $\psi^*$ its convex conjugate.

For our proof we will operate in the finite state-action space, as in the original work (Ho & Ermon, 2016). In this setting, cost functions can be represented as vectors in $\mathbb{R}^{\mathcal{S}\times\mathcal{A}}$, and joint state-action distributions can be represented as vectors in $[0,1]^{\mathcal{S}\times\mathcal{A}}$. Let $f$ be the function defining some $f$-divergence. Given the expert for the task, we can define the following cost function regularizer,

$$\psi_f(c) := \mathbb{E}_{\rho^{\exp}(s,a)}\left[f^*(c(s,a)) - c(s,a)\right] \tag{15}$$

where $f^*$ is the convex conjugate of $f$. Given this choice, with simple algebraic manipulation done in appendix B we have,

$$\psi_f^*(\rho^{\pi}(s,a) - \rho^{\exp}(s,a)) = D_f\left(\rho^{\pi}(s,a)||\rho^{\exp}(s,a)\right) \tag{16}$$

$$\mathrm{RL} \circ \mathrm{IRL}_{\psi}(\pi^{\exp}) = \underset{\pi}{\arg\min} -\mathcal{H}^{\mathrm{causal}}(\pi) + D_f\left(\rho^{\pi}(s,a)||\rho^{\exp}(s,a)\right) \tag{17}$$

Typically, the causal entropy term is considered a policy regularizer, and is weighted by $0 \leq \lambda \leq 1$. Therefore, modulo the term $\mathcal{H}^{\mathrm{causal}}(\pi)$, our derivations show that $f$-MAX, and by inheritance AIRL (Fu et al., 2018), all fall under the cost-regularized Max-Ent IRL framework of Ho & Ermon (2016)!

# 4 UNDERSTANDING THE RELATIONS AMONG LEARNING-FROM-DEMONSTRATION ALGORITHMS

Given results derived in the prior section, we can now begin to populate table 1, writing various Imitation Learning algorithms in a common form, as the minimization of some statistical divergence between $\rho^{\exp}(s,a)$ and $\rho^{\pi}(s,a)$. In Behaviour Cloning we minimize $\mathbb{E}_{\rho^{\exp}(s)}\left[\mathrm{KL}\left(\rho^{\exp}(a|s)||\rho^{\pi}(a|s)\right)\right]$[5]. On the other hand, the corollary in section 3.1 demonstrates that AIRL (Fu et al., 2018) minimizes $\mathrm{KL}\left(\rho^{\pi}(s,a)||\rho^{\exp}(s,a)\right)$, while GAIL (Ho & Ermon, 2016) optimizes $D_{JS}(\rho^{\exp}(s,a)||\rho^{\pi}(s,a)) - \lambda\mathcal{H}^{\mathrm{causal}}(\pi)$. Hence, there are two ways in which the direct IRL methods differ from BC. First, in standard BC the policy is optimized to match the conditional distribution $\rho^{\exp}(a|s)$, whereas in the other two the policy is explicitly encouraged to match the marginal state distributions as well. Second, in BC we make use of the forward KL divergence, whereas AIRL and GAIL use divergences that exhibit more mode-seeking behaviour. These observations allow us to generate the following two hypotheses about why direct IRL methods outperform BC, particularly in the low-data regime,

---

[5]Since it is equal to minimizing $-\mathbb{E}_{\rho^{\exp}(s,a)}\left[\log \rho^{\pi}(a|s)\right] - \mathcal{H}^{\exp}(s,a)$ and $\mathcal{H}^{\exp}(s,a)$ is constant w.r.t. the policy ($\mathcal{H}^{\exp}(s,a)$ is the entropy of $\rho^{\exp}(s,a)$)

> **Hypothesis 1** *In common MDPs of interest, the reward function depends more on the state than the action. Hence it is plausible that matching state marginals is more useful than matching action conditional marginals.*

> **Hypothesis 2** *It is known that optimization using the forward KL divergence results in distributions with a mode-covering behaviour, whereas using the reverse KL results in mode-seeking behaviour (Bishop, 2006). Therefore, since in Reinforcement Learning we care about the "quality of trajectories", being mode-seeking is more beneficial than mode-covering, particularly in the low-data regime.*

In what follows, we seek to experimentally evaluate our hypotheses. To tease apart the differences between the direct Max-Ent IRL methods and BC, we present an algorithm that optimizes $\mathrm{KL}\left(\rho^{\exp}(s,a)||\rho^\pi(s,a)\right)$. We then compare its performance to Behaviour Cloning and the standard AIRL algorithm using varying amounts of expert demonstrations.

## 5 DERIVING FORWARD KL

While $f$-MAX is a general algorithm, useful for most choices of $f$, it unfortunately cannot be used for the special case of forward KL, i.e. $\mathrm{KL}\left(\rho^{\exp}(s,a)||\rho^\pi(s,a)\right)$. In the following sections we identify the problem and present a separate direct Max-Ent IRL method that optimizes this divergence.

### 5.1 DERIVING FORWARD KL FROM $f$-MAX

Let $T_\omega^\pi$ denote the maximizer of equation 8 for a given policy $\pi$. For the case of forward KL, drawing upon equations from Nowozin et al. (2016) we have,

$$u := \frac{\rho^{\exp}(s,a)}{\rho^\pi(s,a)} \qquad f(u) := u\log u \qquad f^*(t) = \exp(t-1) \qquad T_\omega^\pi = 1 + \log\frac{\rho^{\exp}(s,a)}{\rho^\pi(s,a)} \tag{18}$$

Given this, the objective for the policy (equation 9) under the optimal $T_\omega^\pi$ becomes,

$$\max_\pi \mathbb{E}_{\tau\sim\pi}\left[\sum_t f^*(T_\omega^\pi(s_t,a_t))\right] \propto \mathbb{E}_{(s,a)\sim\rho^\pi(s,a)}\left[f^*(T_\omega^\pi(s,a))\right] \tag{19}$$

$$= \mathbb{E}_{(s,a)\sim\rho^\pi(s,a)}\left[\exp\left(\left(1+\log\frac{\rho^{\exp}(s,a)}{\rho^\pi(s,a)}\right)-1\right)\right] \tag{20}$$

$$= \mathbb{E}_{(s,a)\sim\rho^\pi(s,a)}\left[\frac{\rho^{\exp}(s,a)}{\rho^\pi(s,a)}\right] \tag{21}$$

$$= 1 \tag{22}$$

Hence, there is no signal to train the policy! [6]

### 5.2 AN ALTERNATIVE METHOD FOR FORWARD KL

In this section we derive an algorithm for optimizing $\mathrm{KL}\left(\rho^{\exp}(s,a)||\rho^\pi(s,a)\right)$. Similar to AIRL (Fu et al., 2018), let us have a discriminator, $D(s,a)$ whose objective is to discriminate between expert and policy state-action pairs,

$$\max_D \mathbb{E}_{(s,a)\sim\rho^{\exp}(s,a)}\left[\log D(s,a)\right] + \mathbb{E}_{(s,a)\sim\rho^\pi(s,a)}\left[\log\left(1-D(s,a)\right)\right] \tag{23}$$

---

[6]A similar results holds for the standard $f$-GAN formulation (Nowozin et al., 2016).

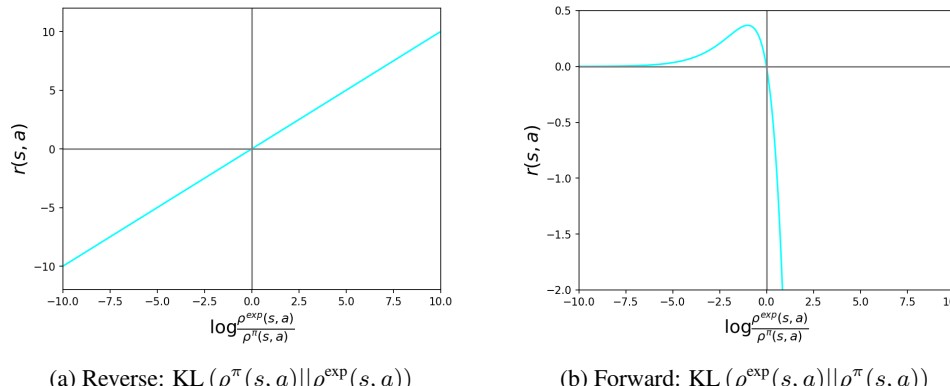

(a) Reverse: KL $\left(\rho^\pi(s,a)||\rho^{\exp}(s,a)\right)$        (b) Forward: KL $\left(\rho^{\exp}(s,a)||\rho^\pi(s,a)\right)$

Figure 1: $r(s,a)$ as the function of the logits of the optimal discriminator, $\ell^\pi(s,a) = \log \frac{\rho^{\exp}(s,a)}{\rho^\pi(s,a)}$.

We now define the objective for the policy to be,

$$h(s,a) := \log D(s,a) - \log\left(1 - D(s,a)\right) \tag{24}$$

$$r(s,a) := \exp(h(s,a)) \cdot (-h(s,a)) \tag{25}$$

$$\max_\pi \mathbb{E}_{\tau \sim \pi_\theta}\left[\sum_t r(s_t, a_t)\right] \tag{26}$$

In appendix C we show,

$$\mathbb{E}_{\tau \sim \pi}\left[\sum_t r(s_t, a_t)\right] \propto -\mathrm{KL}(\rho^{\exp}(s,a)||\rho^\pi(s,a)) \tag{27}$$

This is a refreshing result since it demonstrates that we can convert the AIRL algorithm (Fu et al., 2018) into its forward KL counterpart by simply modifying the reward function used; in AIRL (reverse KL) the reward is defined as $r(s,a) := \log D(s,a) - \log\left(1 - D(s,a)\right)$, whereas for forward KL it is defined as $r(s,a) := \frac{D(s,a)}{1-D(s,a)} \cdot \log \frac{1-D(s,a)}{D(s,a)}$. We refer to this forward KL version of AIRL as FAIRL.

If we parameterize the discriminator as $D(s,a) := \sigma(\ell(s,a))$, where $\sigma$ represents the sigmoid activation function, the logit of the discriminator, $\ell(s,a)$, is equal to $\log D(s,a) - \log\left(1 - D(s,a)\right)$. Hence, for an optimal discriminator, $D^\pi$, we have $\ell^\pi(s,a) = \log \frac{\rho^{\exp}(s,a)}{\rho^\pi(s,a)}$. It is instructive to plot the reward functions under the two different settings as a function of $\ell^\pi(s,a)$; figure 1 presents these plots. As can be seen, in the forward KL version of AIRL, if for a state-action pair the expert puts more probability mass than the policy, the policy is *severely* punished. However, if for some state-action pairs the policy places a lot more mass than the expert, it almost does not matter. As a result, the policy spreads its mass. On the other hand, in the original AIRL formulation (reverse KL), the policy is *always* encouraged to put less mass than the expert. These observations are in line with standard intuitions about the mode-covering/mode-seeking behaviours of the two KL divergences (Bishop, 2006).

## 6 EXPERIMENTS

In this section we provide empirical comparisons between AIRL, FAIRL, and standard BC in the Ant and Halfcheetah environments found in Open-AI Gym (Brockman et al., 2016).

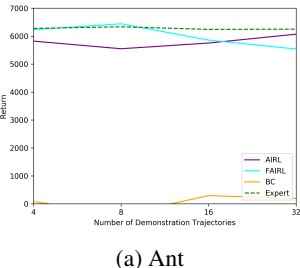

(a) Ant

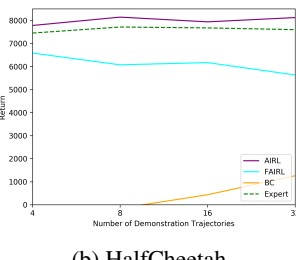

(b) HalfCheetah

Figure 2: Average return on 50 evaluation trajectories as a function of number of expert demonstrations (higher is better). Models evaluated *deterministically*. As we ran two seeds per experiment, we do not present standard deviations. While FAIRL performs comparably to AIRL, Behaviour cloning lags behind quite significantly. Considering the form of their objectives (table 1), this demonstrates that the advantage of direct Max-Ent IRL methods over BC is a result of the additional aspect of their objectives explicitly matching marginal state distributions.

## 6.1 SETUP

**Expert Policy**   To simulate access to expert demonstrations we train an expert policy using Soft-Actor-Critic (SAC) (Haarnoja et al., 2018), a state-of-the-art reinforcement learning algorithm for continuous control. The expert policy consists of a 2-layer MLP with 256-dim layers, ReLU activations, and two output streams for the mean and the diagonal covariance of a `Tanh(Normal(`$\mu, \sigma$`))` distribution [7]. We use the default hyperparameter settings for training the expert.

**Evaluation Setup**   Using a trained expert policy, we generated 4 sets of expert demonstrations of that contain $\{4, 8, 16, 32\}$ trajectories. Starting from a random offset, each trajectory is subsampled by a factor of 20. This is standard protocol employed in prior direct methods for Max-Ent IRL (Ho & Ermon, 2016; Fu et al., 2018). Also note that when generating demonstrations we *sample* from the expert's action distribution rather than taking the mode. This way, since the expert was trained using Soft-Actor-Critic, the expert should correspond to the Max-Ent optimal policy for the reward function $\frac{1}{\tau} r_g(s, a)$, where $\tau$ is the SAC temperature used and $r_g(s, a)$ is the ground-truth reward function. To compare the various learning-from-demonstration algorithms we train each method at each amount of expert demonstrations using 2 random seeds. For each seed, we checkpoint the model at its best validation loss[8] throughout training. At the end of training, the resulting checkpoints are evaluated on 50 test episodes.

**Details for AIRL & FAIRL**   For AIRL and FAIRL, the student policy has an identical architecture to that of the expert, and the discriminator is a 2-layer MLP with 256-dim layers and Tanh activations. We normalize the observations from the environment by computing the mean and standard deviations of the expert demonstrations. The RL algorithm used for the student policies is SAC (Haarnoja et al., 2018), and the temperature parameter is tuned separately for AIRL & FAIRL.

**Details for BC**   For BC, we use an identical architecture as the expert. The model was fit using Maximum Likelihood Estimation[9]. As before, the observations from the environment are normalized using the mean and standard deviation of the expert demonstrations.

## 6.2 RESULTS & INTERPRETATIONS

To match state-action marginals, the optimal student policy must *sample* actions from the state-conditional distribution, $\pi(a|s)$. On the other hand, when we deploy a trained policy it is reasonable to instead choose the mode of this distribution, which we call the deterministic setting. Here, we present evaluation results under the former setting, and defer the results for the deterministic setting to the appendix.

---

[7]This is the architecture presented in SAC (Haarnoja et al., 2018)

[8]Average return on 10 test episodes

[9]Recall that given a state, the output of the policy is a `Tanh(Normal(`$\mu, \sigma$`))` distribution

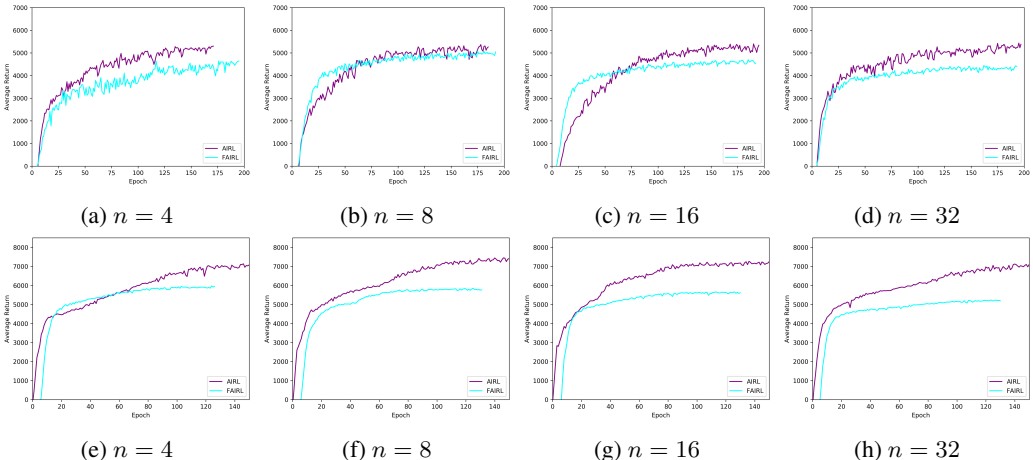

Figure 3: Validation curves throughout training using stochastic evaluation (refer to appendix D for deterministic evaluation results). Top row Ant, Bottom row Halfcheetah. $n$ represents the number of expert demonstrations provided. Due to its mode-covering behaviour, FAIRL does not perform as well as AIRL when evaluated stochastically. However, with determinisitc evaluation FAIRL outperforms AIRL in the Ant environment.

Figure 4 demonstrates that both AIRL and FAIRL outperform BC by a large margin, especially in the low data regime. Specifically, the fact that FAIRL outperforms BC supports hypothesis 1 that the performance gain of Max-Ent IRL is not necessarily due to the direction of KL divergence used, but is the result of explicitly encouraging the policy to match the marginal state distribution of the expert in addition to the matching of conditional action distribution.

To compare AIRL and FAIRL, in figure 3 we plot the validation curves throughout training using stochastic evaluation. Across the two tasks and various number of expert demonstrations, AIRL consistently outperforms FAIRL. When using deterministic evaluation (figure 5), FAIRL achieves a significant performance gain to the point that it outperforms AIRL on the Ant environment across all demonstrations set sizes. Such observations provide initial positive support for hypothesis 2; as more expert demonstrations are provided, the policy trained with FAIRL broadens its distribution to cover the data-distribution, resulting in trajectories accumulating less reward in expectation. We note however that more detailed experiments are necessary for adequately comparing the two methods.

## 7 CONCLUSION & FUTURE WORK

The motivation for this work stemmed from the superior performance of recent direct Max-Ent IRL methods (Ho & Ermon, 2016; Fu et al., 2018) compared to BC in the low-data regime, and the desire to understand the relation between various approaches for Learning from Demonstrations. We first presented $f$-MAX, a generalization of AIRL (Fu et al., 2018), which allowed us to interpret AIRL as optimizing for KL $(\rho^\pi(s,a)||\rho^{\exp}(s,a))$. We demonstrated that $f$-MAX, and by inhertance AIRL, is a subset of the cost-regularized IRL framework laid out by Ho & Ermon (2016). Comparing to the standard BC objective, $\mathbb{E}_{\rho^{\exp}(s)}[\text{KL}(\rho^{\exp}(a|s)||\rho^\pi(a|s))]$, we hypothesized two reasons for the superior performance of AIRL: 1) the additional terms in the objective encouraging the matching of marginal state distributions, and 2) the direction of the KL divergence being optimized. Setting out to empirically evaluate these claims we presented FAIRL, a one-line modification of the AIRL algorithm that optimizes KL $(\rho^{\exp}(s,a)||\rho^\pi(s,a))$. FAIRL outperformed BC in a similar fashion to AIRL, which allowed us to conclude the key factor being the matching of state marginals. Additional comparisons between FAIRL and AIRL provided initial understanding about the role of the direction of the KL being optimized. In future work we aim to produce results on a more diverse set of more challenging environments. Additionally, evaluating other choices of $f$-divergence beyond forward and reverse KL may present interesting avenues for improvement (Wang et al., 2018). Lastly, but importantly, we would like to understand whether the mode-covering behaviour of FAIRL could result in more robust policies (Rajeswaran et al., 2017).

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

## A    SOME USEFUL IDENTITIES

Let $h : \mathcal{S} \times \mathcal{A} \to \mathbb{R}$ be an arbitrary function. If all episodes have the same length $T$, we have,

$$\mathbb{E}_{\tau \sim \pi} \left[ \sum_t h(s_t, a_t) \right] = \sum_t \mathbb{E}_{(s_t, a_t) \sim \rho^\pi(s_t, a_t)} \left[ h(s_t, a_t) \right] \tag{28}$$

$$= \sum_t \int_{S,A} \rho^\pi(s_t, a_t) h(s_t, a_t) \tag{29}$$

$$= \int_{S,A} \left[ \sum_t \rho^\pi(s_t, a_t) \right] h(s, a) \tag{30}$$

$$= T \cdot \int_{S,A} \rho^\pi(s, a) h(s, a) \tag{31}$$

$$= T \cdot \mathbb{E}_{(s,a) \sim \rho^\pi(s,a)} \left[ h(s, a) \right] \tag{32}$$

In a somewhat similar fashion, in the infinite horizon case with fixed probability $\gamma \in (0, 1)$ of transitioning to a terminal state, for the discounted sum below we have,

$$\mathbb{E}_{\tau \sim \pi} \left[ \sum_t \gamma^t h(s_t, a_t) \right] = \sum_t \mathbb{E}_{(s_t, a_t) \sim \rho^\pi(s_t, a_t)} \left[ \gamma^t h(s_t, a_t) \right] \tag{33}$$

$$= \sum_t \int_{S,A} \gamma^t \rho^\pi(s_t, a_t) h(s_t, a_t) \tag{34}$$

$$= \int_{S,A} \left[ \sum_t \gamma^t \rho^\pi(s_t, a_t) \right] h(s, a) \tag{35}$$

$$= \Gamma \cdot \int_{S,A} \rho^\pi(s, a) h(s, a) \tag{36}$$

$$= \Gamma \cdot \mathbb{E}_{(s,a) \sim \rho^\pi(s,a)} \left[ h(s, a) \right] \tag{37}$$

where $\Gamma := \frac{1}{1-\gamma}$ is the normalizer of the sum $\sum_t \gamma^t$. Since the integral of an infinite series is not always equal to the infinite series of integrals, some analytic considerations must be made to go from equation 34 to 35. But, one simple case in which it holds is when the ranges of $h$ and all $\rho^\pi(s_t, a_t)$ are bounded.

## B    SIMPLE ALGEBRAIC MANIPULATION

$$\psi_f^*(\rho^\pi(s, a) - \rho^{\exp}(s, a)) = \sup_{c \in \mathbb{R}^{\mathcal{S} \times \mathcal{A}}} \left[ (\rho^\pi(s, a) - \rho^{\exp}(s, a))^T c \quad - \quad \psi_f(c) \right] \tag{38}$$

$$= \sup_{c \in \mathbb{R}^{\mathcal{S} \times \mathcal{A}}} \left[ \sum_{\mathcal{S} \times \mathcal{A}} (\rho^\pi(s, a) - \rho^{\exp}(s, a)) \cdot c(s, a) \quad - \quad \sum_{\mathcal{S} \times \mathcal{A}} \rho^{\exp}(s, a) \cdot (f^*(c(s, a)) - c(s, a)) \right] \tag{39}$$

$$= \sup_{c \in \mathbb{R}^{\mathcal{S} \times \mathcal{A}}} \left[ \sum_{\mathcal{S} \times \mathcal{A}} [\rho^\pi(s, a) \cdot c(s, a) - \rho^{\exp}(s, a) \cdot f^*(c(s, a))] \right] \tag{40}$$

$$= \sup_{c \in \mathbb{R}^{\mathcal{S} \times \mathcal{A}}} \left[ \mathbb{E}_{\rho^\pi(s,a)} [c(s, a)] - \mathbb{E}_{\rho^{\exp}(s,a)} [f^*(c(s, a))] \right] \tag{41}$$

$$= \sup_{T_\omega \in \mathbb{R}^{\mathcal{S} \times \mathcal{A}}} \left[ \mathbb{E}_{\rho^\pi(s,a)} [T_\omega(s, a)] - \mathbb{E}_{\rho^{\exp}(s,a)} [f^*(T_\omega(s, a))] \right] \tag{42}$$

$$= D_f \left( \rho^\pi(s, a) || \rho^{\exp}(s, a) \right) \tag{43}$$

To go from 41 to 42 we simply changed notation $T_\omega(s, a) := c(s, a)$, and we can go from 42 to 43 because it is the exact same form as the variational characterization of $f$-divergences shown in equation **??**. Note that equation 42 suggests the same training procedure as described for $f$-MAX.

## C  DERIVATION FOR FAIRL

Below we present the derivation for equation 27. Recalling definitions,

$$h(s, a) := \log D(s, a) - \log (1 - D(s, a)) \tag{44}$$
$$r(s, a) := \exp(h(s, a)) \cdot (-h(s, a)) \tag{45}$$

and assuming the discriminator is optimal[10], we have,

$$\mathbb{E}_{\tau \sim \pi} \left[ \sum_t r(s_t, a_t) \right] = \mathbb{E}_{\tau \sim \pi} \left[ \sum_t \exp(h(s_t, a_t)) \cdot (-h(s_t, a_t)) \right] \tag{46}$$

$$= \mathbb{E}_{\tau \sim \pi} \left[ \sum_t \frac{\rho^{\exp}(s_t, a_t)}{\rho^\pi(s_t, a_t)} \cdot \log \frac{\rho^\pi(s_t, a_t)}{\rho^{\exp}(s_t, a_t)} \right] \tag{47}$$

$$\propto \mathbb{E}_{(s,a) \sim \rho^\pi(s,a)} \left[ \frac{\rho^{\exp}(s_t, a_t)}{\rho^\pi(s_t, a_t)} \cdot \log \frac{\rho^\pi(s_t, a_t)}{\rho^{\exp}(s_t, a_t)} \right] \tag{48}$$

$$= \mathbb{E}_{(s,a) \sim \rho^{\exp}(s,a)} \left[ \log \frac{\rho^\pi(s_t, a_t)}{\rho^{\exp}(s_t, a_t)} \right] \tag{49}$$

$$= -\text{KL}(\rho^{\exp}(s, a) || \rho^\pi(s, a)) \tag{50}$$

## D  DETERMINISTIC EVALUATION

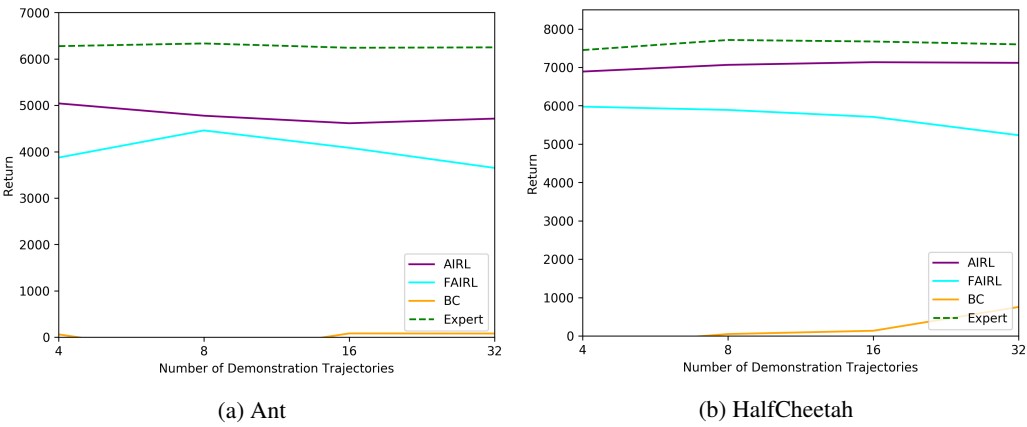

(a) Ant

(b) HalfCheetah

Figure 4: Average return on 50 evaluation trajectories as a function of number of expert demonstrations (higher is better). Models evaluated *stochastically*. As we ran two seeds per experiment, we do not present standard deviations. While FAIRL performs comparably to AIRL, Behaviour cloning lags behind quite significantly. Considering the form of their objectives (table 1), this demonstrates that the advantage of direct Max-Ent IRL methods over BC is a result of the additional aspect of their objectives explicitly matching marginal state distributions.

---

[10]As a reminder, the optimal discriminator has the form, $D(s, a) = \frac{\rho^{\exp}(s,a)}{\rho^{\exp}(s,a) + \rho^\pi(s,a)}$. A simple proof of which can be found in Goodfellow et al. (2014).

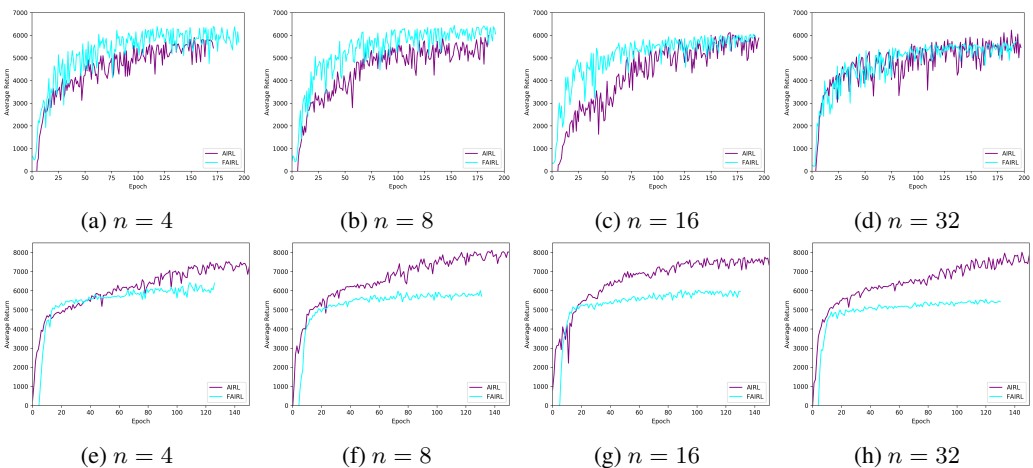

(a) $n = 4$    (b) $n = 8$    (c) $n = 16$    (d) $n = 32$

(e) $n = 4$    (f) $n = 8$    (g) $n = 16$    (h) $n = 32$

Figure 5: Training curves averaged across random seeds. Top row Ant, Bottom row Halfcheetah. $n$ represents the number of expert demonstrations provided. While in the Ant environment FAIRL has slight performance gains, in Halfcheetah AIRL performs noticebaly better. Further experiments are necessary to compare these two methods.

