# OpenReview forum: "Understanding the Relation Between Maximum-Entropy Inverse Reinforcement Learning and Behaviour Cloning"
_ICLR.cc/2019/Workshop/DeepGenStruct — DeepGenStruct 2019_

### Official Review · AnonReviewer1 · 2019-04-17
**Maximum-entropy inverse RL methods and comparison with Behaviour Cloning**

**Rating:** 4
**Confidence:** 2

**Review:**

The paper presents a framework to unify different Maximum-entropy inverse RL methods and to compare them with Behaviour Cloning (BC). Specifically, the paper is motivated by the question of why BC performs significantly worse than Maximum-Entropy IRL in the small data regime. The authors hypothesise that the reasons are (i) that BC tries to model conditional policies while other approaches try to match also state marginals and (ii) due to the moment matching properties of BC as opposed to  mode seeking divergences that can better match expert policies to learning policies.

I believe that the paper presents an interesting unified framework based on f-divergences together with some novel modifications of previous methods. The experiments on continuous optimal control support to some extend the theoretical analysis of the paper. However, a more extensive experimental comparison is needed to draw more clear conclusions.

---

### Official Review · AnonReviewer2 · 2019-04-17

**Rating:** 3
**Confidence:** 2

**Review:**

This paper discusses several adversarial imitation learning algorithms and connects them through f-divergence. A variant of the algorithms is proposed that minimizes forward KLD b/w the expert policy and the student policy.

The f-divergence formulations of adversarial IL in the paper are directly derived from the f-GAN work, which is thus pretty straightforward.

The two hypotheses that matching state marginals (instead of action marginals) is better and that forward KL is better than reverse KL look reasonable.

---

### Decision · Program_Chairs · 2019-04-19
**Acceptance Decision**

Accept